# Global estimates of ambient reactive nitrogen components during 2000-2100 based on the multi-stage model

Rui Li[a, b *], Yining Gao[a], Lijia Zhang[a], Yubing Shen[a], Tianzhao Xu[c], Wenwen Sun[c *], Gehui Wang[a]

[a] *Key Laboratory of Geographic Information Science of the Ministry of Education, School of Geographic Sciences, East China Normal University, Shanghai, 200241, PR China*

[b] *Institute of Eco-Chongming (IEC), 20 Cuiniao Road, Chenjia Town, Chongming District, Shanghai, 202162, China*

[c] *Department of Research, Shanghai University of Medicine & Health Sciences Affiliated Zhoupu Hospital, Shanghai 201318, China*

**\* Corresponding author**

Prof. Li (rli@geo.ecnu.edu.cn) and Prof. Sun (sunww@sumhs.edu.cn)

**Abstract**

High contents of reactive nitrogen components aggravate air pollution and could also impact ecosystem structure and function across the terrestrial-aquatic-marine continuum. However, the long-term historical trends and future prediction of reactive nitrogen components at the global scale still remains high uncertainties. In our study, the field observations, satellite products, model output, and many other covariates were integrated into the multi-stage machine-learning model to capture the global patterns of reactive nitrogen components during 2000-2019. In order to decrease the estimate uncertainties in the future scenarios, the constructed reactive nitrogen component dataset during the historical period was then utilized as the constraint to calibrate the CMIP6 dataset in four scenarios. The results suggested the cross-validation (CV) $R^2$ values of four species showed satisfied performance ($R^2 > 0.55$). The concentrations of estimated reactive nitrogen components in China experienced persistent increases during 2000-2013, while they suffered from drastic decreases since 2013 except $NH_3$. It might be associated with the impact of clean air policy. However, these compounds in Europe and the United States remained relatively stable since 2000. In the future scenarios, SSP3-7.0 (traditional energy scenario) and SSP1-2.6 (carbon neutrality scenario) showed the highest and lowest reactive nitrogen component concentrations, respectively. Although the reactive nitrogen concentrations in some heavy-pollution scenarios (SSP3-7.0) also experienced decreases during 2020-2100, SSP1-2.6 and SSP2-4.5 (middle emission scenario) still kept more

rapid decreasing trends. Our results emphasize the need for carbon-neutrality pathway to reduce
global atmospheric N pollution.

## 1. Introduction

Along with the development of global urbanization and industrialization, the anthropogenic
emissions of reactive nitrogen (e.g., $NO_x$, $NH_3$) experienced drastic increases during the past
decades, and caused the higher concentrations of $NO_2$, $NH_3$, and many secondary components such
as $NO_3^-$ ($NO_3$-N), $NH_4^+$ ($NH_4$-N), and $HNO_3$ (Chen et al., 2021; Liu et al., 2020b; McDuffie et al.,
2020). The reactive nitrogen released from anthropogenic source could significantly alter the global
nitrogen cycle throughout the Earth system (Altieri et al., 2021; Zhang et al., 2020). Reactive
nitrogen in the atmosphere dominates the chemical formation of tropospheric $O_3$ and aggravates the
particle pollution (Geddes and Martin, 2017), with implications for global air quality and climate
change (He et al., 2022; Von Schneidemesser et al., 2015). Moreover, the ambient reactive nitrogen
could be deposited into the land surface and could cause lake eutrophication and soil acidification
(Bouwman et al., 2002; Chen et al., 2018). Therefore, it is highly necessary to understand the spatial
distributions and temporal evolution trends of reactive nitrogen components at the global scale.
Despite the global importance, observational constraints on reactive nitrogen in the atmosphere
were still scarce in most parts of the world (Liu et al., 2020b). Furthermore, the majority of
monitoring sites focused on China, Europe, and the United States (Du et al., 2014; Li et al., 2020;
Li et al., 2019a; Li et al., 2016), and these uneven sites only possessed limited spatial
representativeness (Shi et al., 2018), which restricted the accurate assessment of global reactive
nitrogen pollution. Fortunately, the satellite observations gave us a unprecedent chance to capture
the global variations of atmospheric reactive nitrogen. Geddes et al. (2017) used the satellite
products to calibrate the simulated reactive nitrogen oxides ($NO_y$) and improved the predictive
performance (R = 0.83) compared with the chemical transport model (CTM) output alone (Geddes
and Martin, 2017). Afterwards, Liu et al. (2022) also used the similar method to estimate the global
wet deposition of reduced nitrogen ($NH_4^+$) and the R value achieved 0.80(Liu et al., 2021). Although
the calibration based on satellite products could improve the predictive accuracy compared with
CTM output, the simulated values still largely biased from the ground-level observations. Moreover,
the method cannot accurately fill the gaps of reactive nitrogen concentrations without satellite

coverage. In our previous works, we developed a satellite-based ensemble machine-learning model to predict the wet $NH_4^+$ deposition across China and the $R^2$ value reached 0.76 (R = 0.88) (Li et al., 2020). However, this technique was not expanded to the global scale and the high-accuracy and full-coverage global ambient reactive nitrogen dataset was still lack.

Apart from the historical estimates, the future prediction of reactive nitrogen is also important because these components in the future scenarios could significantly affect the land carbon cycle and greenhouse gas emissions, both of which could aggravate the global climate change and affect the earth system safety (Chen et al., 2015; Zaehle, 2013). To the best of our knowledge, only two studies focused on global aerosol prediction in the future scenarios. Chen et al. (2023) predicted the global $PM_{2.5}$ levels and associated mortalities in 2100 under different climate scenarios and found that SSP3-7.0 scenario was linked with the highest $PM_{2.5}$ exposure. Li et al., (2022) also simulated the global $NO_3^-$ ($NO_3$-N) and $NH_4^+$ ($NH_4$-N) levels in the future four scenarios and demonstrated that both of these components showed marked decreases in most cases except SSP5-8.5 scenario. However, this study predicted the future reactive nitrogen based on historical CTM output alone, which lacks of observation constraints. The result might increase the uncertainty of assessment.

In our study, we developed a multi-stage model to estimate the concentrations of four reactive nitrogen species ($NO_3^-$ ($NO_3$-N), $HNO_3$, $NH_3$, and $NH_4^+$ ($NH_4$-N)) during 2000-2019 because these species were most important reactive nitrogen components for human health and ecological ecosystem and also showed abundant ground-level observations. Then, the species over the 2020-2100 period under the SSP1-2.6, SSP2-4.5, SSP3-7.0, and SSP5-8.5 scenarios were also corrected based on the historical estimates. Finally, the long-term dataset of reactive nitrogen during 2000-2100 was constructed. Our results were beneficial to assess the impacts of reactive nitrogen components on air pollution and climate change in the future.

**2. Material and methods**

2.1 Reactive nitrogen observations

Most of reactive nitrogen observations focused on East Asia, Europe, and the United States. The monthly reactive nitrogen components monitoring data during 2010-2015 in China were downloaded from nationwide nitrogen deposition monitoring network (NNDMN) including 32 sites, and these sites could be classified into three types mixed with urban, rural, and background sites

(Xu et al., 2019) (Table S1). The concentrations of reactive nitrogen components were determined
using the active DEnuder for Long-Term Atmospheric sampling system (DELTA). The detailed
sampling and analysis procedures have been described by Xu et al. (2019). The dataset of reactive
nitrogen components in other countries of East Asia during 2000-2019 were download from the
Acid Deposition Monitoring Network in East Asia (EANET), which includes 41 sites. The European
Monitoring and Evaluation Programme (EMEP) provides records of long-term reactive nitrogen
components in 86 sites of most countries across West Europe. Monthly reactive nitrogen
components dataset in 84 locations across the United States could be obtained from the Clean Air
Status and Trends Network (CASTNET) (Figure S1).
2.2 Data preparation
The GEOS-Chem (v13.4.0) model driven by MERRA2 meteorological parameters was applied
to simulate the historical reactive nitrogen components (daily) during 2000-2019 (Feng et al., 2021).
The GEOS-Chem model was composed of detailed ozone-$NO_x$-VOC-PM-halogen tropospheric
chemistry. The grid version of the model with a horizontal resolution of $2° \times 2.5°$ was utilized. Wet
deposition contained many processes including sub-grid scavenging in convective updrafts, in-
cloud rainout, and below-cloud washout (Liu et al., 2001). Dry deposition was estimated based on
a resistance-in-series model (Wesely, 2007). The estimates of aerosol optical properties account for
the hygroscopic growth (Drury et al., 2010). Vertical mixing in the boundary layer follows a non-
local scheme implemented by Lin and McElroy (2010), and convection employs the relaxed
Arakawa-Schubert scheme. The anthropogenic emission inventory in 2000-2019 was downloaded
from the website of Community Emissions Data System (CEDS) (Hoesly et al., 2018). CEDS
emission inventory includes eight sectors such as agriculture, energy, industry, residential, shipping,
solvents, transportation, and waste incineration. Then, the daily reactive nitrogen components were
averaged to the monthly scale.
The IASI instrument aboard on the polar sun-synchronous MetOp platform traverses the
equation twice each day (9:30 a.m. and 9:30 p.m. local solar time) (Whitburn et al., 2016a). The
measurements in the daytime usually shows the better accuracy than those at night due to the high
sensitivity to ambient $NH_3$ (Van Damme et al., 2017; Whitburn et al., 2016a; Whitburn et al., 2016b).
In our study, we used the IASI $NH_3$ columns in morning during 2008-2019 to estimate the $NH_3$ and
$NH_4^+$ concentrations globally. Besides, the $NH_3$ column dataset with a cloud shield higher than 25%
and relative error above 100% were eliminated.
The tropospheric vertical column density (VCD) of $NO_2$ retrieved from OMI aboard on Aura
satellite crosses the earth once a day (Kim et al., 2016). OMI-derived tropospheric $NO_2$ column
densities during 2005-2019 was applied to develop the model. The tropospheric $NO_2$ column density
data with cloud radiance fraction > 0.5, terrain reflectivity > 30%, and solar zenith angles > 85°
were screened (Cooper et al., 2022). Additionally, the $NO_2$ columns from GOME (1995-2003),
SCIAMACHY (2002-2011) and GOME-2 (2007-) were also collected to simulate the $NO_3^-$ ($NO_3$-
N) and $HNO_3$ levels. The similar overpass time of these three instruments (from about 09:30 to
10:30 LT, local time) facilitates the simultaneous use to capture consistent long-term coverage.
However, the dataset cannot cover the $NO_2$ columns since 2017. To overcome the inconsistency of
these satellite products, we applied the linear regression technique to construct the relationship
between OMI-$NO_2$ columns and GOME/SCIAMACHY $NO_2$ columns. The results suggested these
satellite products showed good relationship ($R^2 > 0.6$). At last, the long-term (2000-2019) $NO_2$
columns at the global scale were constructed.
The monthly meteorological parameters derived from ERA-5 comprise of 2 m dewpoint
temperature ($D_{2m}$), 2 m temperature ($T_{2m}$), surface pressure ($S_p$), and total precipitation ($T_p$), 10 m
U wind component (U10), and 10 m V wind component (V10). The population density data during
2000-2020         around         the         world         were         downloaded         from
https://hub.worldpop.org/geodata/listing?id=64. The elevation data was extracted from ETOPO at a
spatial resolution of 1' (Amante and Eakins, 2009) (https://rda.ucar.edu/datasets/ds759.4/). In
addition, the land use types including cropland, forest, grassland, shrubland, tundra, barren land,
and snow/ice were obtained from Liu et al. (2020a). Besides, the CMIP6 dataset in four scenarios
were also applied to predict the reactive nitrogen concentrations during 2020-2100. The dataset
includes 2-m air temperatures, wind speed at 850 and 500 hPa, total cloud cover, precipitation,
relative humidity, and short-wave radiation. The modelled meteorological parameters derived from
16 earth system models were incorporated into the machine-learning model. The detailed models
are summarized in Table S2.
2.3  Model development
A three-stage model was established to capture the full-coverage reactive nitrogen dataset at
the global scale (Figure 1). In the first stage, the ground-level reactive nitrogen species, satellite
products (e.g., OMI-NO$_2$ and IASI-NH$_3$ columns), meteorological parameters, land use types,
population, and simulated reactive nitrogen components derived from GEOS-Chem model were
collected as the independent variables to estimate the gridded reactive nitrogen species at the
period/grid with satellite product based on XGBoost algorithm. In the second stage, the
meteorological parameters, GEOS-Chem output, land use types, and population were applied to fill
the gaps without satellite retrievals. Then, the simulated results based on these models were fused
to obtain the full-coverage reactive nitrogen components and the ground-level observations were
further used to calibrate the full-coverage dataset and the final reactive nitrogen components at the
global scale were simulated. In the last stage, the reactive nitrogen components and meteorological
parameters in four scenarios (SSP1-2.6, SSP2-4.5, SSP3-7.0, and SSP5-8.5) during 2020-2100 were
collected from CMIP6 dataset including 16 earth system models (Table S2). Then, the data in the
future scenarios were integrated into the ensemble model including XGBoost, LightGBM, and
convolutional neural networks (CNN) to further calibrate the modeling results based on historical
dataset (2000-2019) derived from previous two-stage model. The detailed equations of multiple
machine-learning models are summarized as follows:
(1) XGBoost model
$$F^{(t)} = \sum_{i=1}^{n} [l(y_i, \overset{\Lambda}{y}{}^{(t-1)}) + \partial_{y^{(t-1)}} l(y_i, \overset{\Lambda}{y}{}^{(t-1)}) f_t(x_i) + \frac{1}{2} \partial^2_{y^{(t-1)}} l(y_i, \overset{\Lambda}{y}{}^{(t-1)}) f_t^2(x_i)] + \Omega(f_t) \quad (1)$$

where F$^{(t)}$ is the cost function at the t-th period; $\partial$ is the derivative of the function; $\partial^2_{y^{(t-1)}}$
represents the second derivative of the function; $l$ denotes the differentiable convex loss function
that reveals the difference of the predicted value ( $\overset{\Lambda}{y}$ ) of the i-th instance at the t-th period and the
target value (y$_i$); f$_t$(x) represents the increment; $\Omega(f_t)$ reflects the regularizer. Maximum tree
depth and learning rate reached 15 and 0.1, respectively.
(2) LightGBM model
$$\overset{\Lambda}{f} = \arg \min_f E_{y,X} Q(y, f(x)) \quad (2)$$

$$f_T(X) = \sum_{t=1}^{T} f_t(X) \quad (3)$$

where Q(y,f(x)) reflects the a specific loss function; $\sum_{t=1}^{T} f_t(X)$ denotes the regression trees.
Maximum tree depth, learning rate, and feature fraction reached 25, 0.2, and 0.7, respectively.
(3) CNN model
The reactive nitrogen species and meteorological parameters in the future scenarios were applied to
CNN model based on the historical (2000-2019) reactive nitrogen species derived from stage 1-2
model.
$$x \xrightarrow{f:U-Net} y \quad (4)$$

where x ($x_1$, $x_2$,······, $x_n$) represents the reactive nitrogen species and meteorological parameters
derived from CMIP6 dataset; y ($y_1$, $y_2$, ······, $y_n$) denotes the historical (2000-2019) reactive nitrogen
species.
All of the convolution layers showed the same kernel size of 3×3 and used rectified linear unit
(ReLU) as the activation function. Max-pooling layers were employed for adjusting the size of
images to capture better bottleneck information. After each block, the image size could be halved
by using the max pooling layer with kernel size of 2×2, but the number of channels will be doubled.
In our study, the learning rate was set as 0.1 to achieve the best performance.
All of the independent variables collected from multiple sources were resampled to 0.25° grids
using Kriging interpolation. For example, both of the population density and land use types in each
grid were calculated using spatial clipping toolbox. Later on, all of these variables were combined
to develop the model. During the development of multi-stage model, it was highly imperative to
remove some redundant explanatory variables and then determine the optimal variable group. The
redundant variables means that the overall predictive accuracy could degrade after the removal of
these variables.
**3. Results and discussion**
3.1 The modelling performance of historical reactive nitrogen estimates
The multi-stage model was applied to capture the spatiotemporal variations of reactive nitrogen
concentrations during 2000-2100. In our study, we employed XGBoost model to construct the full-
coverage reactive nitrogen dataset during 2000-2020. The cross-validation (CV) $R^2$ values of the
model for $NO_3^-$ ($NO_3$-N), $HNO_3$, $NH_3$, and $NH_4^+$ ($NH_4$-N) estimates reached 0.67, 0.62, 0.58, and
0.60, respectively (Figure 2). RMSE of $NO_3^-$ ($NO_3$-N), $HNO_3$, $NH_3$, and $NH_4^+$ ($NH_4$-N) were 0.55,
0.23, 2.32, and 1.71 $\mu$g N m$^{-3}$, respectively. MAE of $NO_3^-$ ($NO_3$-N), $HNO_3$, $NH_3$, and $NH_4^+$ ($NH_4$-
N) reached 0.19, 0.13, 1.23, and 0.59 $\mu$g N m$^{-3}$. The CV $R^2$ values of $NO_3^-$ ($NO_3$-N), $HNO_3$, and
$NH_4^+$ ($NH_4$-N) estimates were significantly higher than Jia et al. (2016) (0.22, 0.41, and 0.49), while
the CV $R^2$ value of $NO_3^-$ estimate in our study was comparable to Geddes et al. (2017) (0.68)
(Geddes and Martin, 2017). The CV $R^2$ value of $NH_3$ estimates were also close to the results
obtained by Liu et al. (2019) (0.45-0.71) (Liu et al., 2019). Overall, the predictive performances
historical reactive nitrogen was satisfied. Although the CV $R^2$ values in our study were not
significantly higher than those in some previous studies, our study developed the full-coverage (gap-
free) ambient reactive nitrogen dataset, which was superior to some previous studies. Based on the
constructed full-coverage reactive nitrogen dataset, we also developed the ensemble model to
calibrate the CMIP6 dataset in the future scenarios. The CV $R^2$ values of the model for $NO_3^-$ ($NO_3$-
N), $HNO_3$, $NH_3$, and $NH_4^+$ ($NH_4$-N) estimates in the future scenarios reached 0.62, 0.67, 0.56, and
0.60, respectively (Figure S2). RMSE of $NO_3^-$ ($NO_3$-N), $HNO_3$, $NH_3$, and $NH_4^+$ ($NH_4$-N) were 0.58,
0.26, 2.12, and 1.91 $\mu$g N m$^{-3}$, respectively. MAE of $NO_3^-$ ($NO_3$-N), $HNO_3$, $NH_3$, and $NH_4^+$ ($NH_4$-
N) reached 0.22, 0.22, 1.04, and 0.65 $\mu$g N m$^{-3}$. Overall, the ensemble model for these species in the
future scenarios still showed satisfied performance, and thus the result could be treated to be robust.
3.2 The spatial patterns of nitrogen reactive components

219        The global annual mean concentrations of $NO_3^-$, $HNO_3$, $NH_3$, and $NH_4^+$ during 2000-2019

ranged from 0.03 to 9.08, 0.03 to 1.73, 0.21 to 13.9, and 0.08 to 17.1 $\mu$g N m$^{-3}$ with the mean values
of 0.43 $\pm$ 0.24 (standard deviation over grids), 0.28 $\pm$ 0.13, 1.79 $\pm$ 0.85, and 0.65 $\pm$ 0.36 $\mu$g N m$^{-3}$
(Figure S3), respectively. East Asia especially China, West Europe, and the United States obtained
widespread attention due to the developed economy and dense anthropogenic activity.

224        In China, the overall mean ambient $NO_3^-$ ($NO_3$-N), $HNO_3$, $NH_3$, and $NH_4^+$ ($NH_4$-N)

concentrations reached 1.05 $\pm$ 0.62, 0.35 $\pm$ 0.19, 4.05 $\pm$ 1.84, and 2.38 $\pm$ 1.26 $\mu$g N m$^{-3}$, ranging
from 0.07-9.08, 0.06-1.73, 0.84-11.6, and 0.18-13.1 $\mu$g N m$^{-3}$. At the regional scale, the annual mean
$NO_3^-$, $HNO_3$, $NH_3$, and $NH_4^+$ concentrations followed the order of North China Plain (NCP) (4.38,
1.12, 7.22, and 7.69 µg N m$^{-3}$) > Sichuan Basin (2.40 ± 1.01, 0.52 ± 0.28, 4.92 ± 1.71, and 6.02 ±
1.82 µg N m$^{-3}$) (Figure 3). NCP displayed the higher $NO_3^-$ and $HNO_3$ concentrations owing to dense
human activities and strong industry foundation (Qi et al., 2023; Wen et al., 2018), which could emit
a large amount of $NO_x$ to the atmosphere. In both of Yangtze River Delta (YRD) and Pearl River
Delta (PRD), the combustion of fossil fuels and traffic emissions might be the major source of $NO_x$
emission, which aggravated nitrate events via gas-particle conversion processes (Huang et al., 2017;
Li et al., 2017). For Sichuan Basin, the poor topographical or meteorological conditions were major
factors responsible for the severe nitrate pollution (Zhang et al., 2019). It was not surprising that
high ambient $NH_3$ concentrations focused on NCP and Sichuan Basin because many croplands (dry
land) are distributed on these regions (Karra et al., 2021; Potapov et al., 2022), which was the major
source of $NH_3$ emissions with frequent N fertilizer applications (Ma et al., 2022). Besides, N manure
was another major source of $NH_3$ emissions in China, and the percentage of N manure to $NH_3$
emissions exceeds 50% (Kang et al., 2016). The spatial pattern of $NH_4^+$ level was in good agreement
with the $NH_3$ concentration because $NH_4^+$ was often generated from the reaction of $NH_3$ with $SO_2$
and $NO_2$ (Ehrnsperger and Klemm, 2021). Apart from China, many other countries such as South
Korea and Japan also showed the higher ambient reactive nitrogen concentrations. As shown in
Figure 3, the higher reactive N concentrations occurred on the western coasts of South Korea than
on the eastern coasts. The higher reactive N concentrations in Japan mainly focused on the urban
areas around Tokyo, which might be linked with the dense anthropogenic emission in this region
(Li et al., 2024). In Southeast Asia, Indonesia ($NO_3^-$ ($NO_3$-N), $HNO_3$, $NH_3$, and $NH_4^+$ ($NH_4$-N): 0.18,
0.47, 5.72, and 0.44 µg N m$^{-3}$) suffered from the most serious reactive N pollution compared with
other surrounding countries.

250       In Europe, the ambient $NO_3^-$ ($NO_3$-N), $HNO_3$, $NH_3$, and $NH_4^+$ ($NH_4$-N) concentrations ranged

from 0.13 to 2.84, 0.06 to 0.92, 0.35 to 7.81, and 0.22 to 3.77 µg N m$^{-3}$, respectively. The annual
mean $NO_3^-$ ($NO_3$-N), $HNO_3$, $NH_3$, and $NH_4^+$ ($NH_4$-N) levels reached 0.57 ± 0.28, 0.25 ± 0.11, 1.58
± 0.68, and 0.89 ± 0.42 µg N m$^{-3}$, respectively (Figure 4). High concentrations of reactive nitrogen
components focused on the northern part of Italy, central and southern part of Germany, North
France, Poland, and the western part of Russia, which was in good agreement with the spatial pattern
of $NO_x$ and $NH_3$ emissions (Luo et al., 2022; Qu et al., 2020). Emissions Database for Global
Atmospheric Research (EDGAR) suggested that N fertilization and N manure accounted for 43%
and 53% of total $NH_3$ emissions in western Europe (Liu et al., 2019), respectively. Furthermore, Liu
et al. (2019) confirmed that a good relationship between ambient $NH_3$ level and N fertilization plus
N manure (r = 0.62) was observed in Europe. Cooper et al. (2017) employed the inversion model to
estimate $NO_x$ emission in Europe and also found that high $NO_x$ emission was also mainly distributed
on North France, Germany, the northern part of Italy, and Russia, which partly explained the higher
concentrations of reactive nitrogen components in these regions.
In the United States, the ambient $NO_3^-$ ($NO_3$-N), $HNO_3$, $NH_3$, and $NH_4^+$ ($NH_4$-N)
concentrations reached $0.28 \pm 0.12$, $0.19 \pm 0.08$, $2.12 \pm 0.66$, and $0.49 \pm 0.25$ $\mu g$ N $m^{-3}$, with the
range of 0.03-2.35, 0.03-1.31, 0.26-9.96, and 0.10-6.09 $\mu g$ N $m^{-3}$ (Figure 5), respectively. The
hotspots of $NO_3^-$ ($NO_3$-N), $HNO_3$, and $NH_4^+$ ($NH_4$-N) levels focused on the eastern part of the United
States, while the higher $NH_3$ concentration focused on Central Great Plains and some regions in
California such as San Joaquin Valley (6.15 $\mu g$ N $m^{-3}$). Both of bottom-up and top-down $NO_x$ and
$NH_3$ emissions suggested that the spatial distributions of reactive nitrogen components were
strongly dependent on the precursor emissions (McDuffie et al., 2020; Qu et al., 2020).
Besides, some other regions such as India (1.4, 0.5, 6.6, and 4.4 $\mu g$ N $m^{-3}$) especially the
northern part of India (3.1, 0.8, 12.6, and 8.4 $\mu g$ N $m^{-3}$) also experienced severe reactive N pollution
in the atmosphere. Meanwhile, some countries in South America such Brazil and Argentina and in
Africa such as West Africa Coast (Nigeria, Ivory Coast, Ghana, Togo, and Benin) ($HNO_3$ and $NH_3$:
0.3 and 5.0 $\mu g$ N $m^{-3}$) and Democratic Congo (0.4 and 1.6 $\mu g$ N $m^{-3}$) also suffered from serious
$HNO_3$ (Brazil and Argentina: 0.3 and 0.2 $\mu g$ N $m^{-3}$) and $NH_3$ (3.6 and 2.8 $\mu g$ N $m^{-3}$) pollution. The
higher ambient $NH_3$ concentration focused on the northern part of India might be contributed by
two major reasons. First of all, the intensive agricultural activities and high air temperature might
be responsible for the higher $NH_3$ level (Cui, 2023; Wang et al., 2020). Moreover, the relatively low
sulfur dioxide ($SO_2$) and nitrogen oxides ($NO_x$) emissions coupled with high air temperature
restricted the gas-to-particle conversion of $NH_3$ (Wang et al., 2020). The severe $HNO_3$ and $NH_3$
pollution in Brazil, Argentina, and West Africa Coast might be also linked with the dense agricultural
activities (Huneeus et al., 2017).
3.3  The seasonal variations of reactive nitrogen components
The ambient $NO_3^-$ ($NO_3$-N), $HNO_3$, $NH_3$, and $NH_4^+$ ($NH_4$-N) concentrations exhibited
significant seasonal variations (Figure S4-8). $NO_3^-$, $HNO_3$, and $NH_4^+$ displayed the highest and
lowest values in winter (December-February) and summer (June-August), respectively. On the one
hand, the anthropogenic $NO_x$ emission for domestic heating might be higher in winter compared
with other seasons (Lin et al., 2011). On the other hand, the stagnant meteorological conditions
limited the pollutant diffusion (Li et al., 2019b; Liu et al., 2020c). Meanwhile, the higher relative
humidity in winter facilitated the formation of $NH_4NO_3$ (Huang et al., 2016; Xu et al., 2012).
However, both of ambient $NO_3^-$ and $NH_4^+$ concentrations showed the lower concentrations in
summer, which might be attributable to the decomposition of $NH_4NO_3$ under the condition of high
air temperature. In contrast to the secondary inorganic nitrogen, the ambient $NH_3$ level showed the
highest concentration in summer ($1.71 \pm 0.45$ μg N m$^{-3}$). China, Europe, and the United States
suffered from similar $NH_3$ peaks in summer ($4.20 \pm 1.85$, $1.77 \pm 0.65$, and $2.21 \pm 1.04$ μg N m$^{-3}$).
There are two reasons accounting for the fact. At first, mineral N fertilizer or manure application
was mainly performed in spring and early summer (Paulot et al., 2014). Many field observations
have obtained similar $NH_3$ peak in summer (He et al., 2021; Pan et al., 2018). Moreover, summer
often showed the higher air temperature, which promotes the volatilization of ammonium and limits
the gas-to-particle of gaseous $NH_3$ (Liu et al., 2019).
3.4 The historical trends of reactive nitrogen components during 2000-2019
The long-term trends of ambient $NO_3^-$ ($NO_3$-N), $HNO_3$, $NH_3$, and $NH_4^+$ ($NH_4$-N) concentrations
are shown in Figure 6 and Figure S9-12. The $NO_3^-$ concentration in China displayed rapid increase
(9.7%/yr) during 2000-2007, and then it kept the moderate increase (4.2%/yr) during 2007-2013.
However, the $NO_3^-$ ($NO_3$-N) concentration in China experienced the drastic decrease (-2.6%/yr)
since 2013. The ambient $HNO_3$ and $NH_4^+$ ($NH_4$-N) concentrations showed similar trends during this
period. Due to the impact of Clean Air Action, the concentrations of gaseous precursors (e.g., $SO_2$
and $NO_x$) suffered from substantial decreases, which could be transformed into nitrate and
ammonium via heterogeneous reactions (Huang et al., 2019). However, the decreasing rates of $NO_3^-$
were still much lower than those of gaseous precursors (Li et al., 2023). On the one hand, it might
be associated with the increased $O_3$ level and enhanced atmospheric oxidation capacity (AOC),
which led to an increase in the photochemical reaction rate of the secondary components (Wang et

al., 2019). On the other hand, strong $SO_2$ emission control under the Clean Air Action allowed more gaseous $NH_3$ to form nitrate. The ambient $NH_3$ level remained relatively stable status during 2000-2013, while it experienced rapid increases after 2013. The result was in good agreement with Liu et al. (2019). In fact, the ambient $NH_3$ level in North China Plain still experienced dramatic increase (> 0.2 $\mu g\ N\ m^{-3}$/yr) during 2000-2013 because enhanced agricultural activities. Zhang et al. (2017) have demonstrated that the livestock manure and fertilizer application generally accounted for 43% and 36% of the agricultural $NH_3$ emission, respectively. Since 2013, the $NH_3$ concentration in the entire China suffered from rapid increase, which might be associated with the drastic decrease of sulfate. It was well known that $NH_3$ could react with $HNO_3$ and gaseous $H_2SO_4$ to generate ammonia sulfate and ammonia nitrate (Wang et al., 2022; Wang et al., 2019). Substantial decreases of acidic gases (e.g., $SO_2$) lead to the reduction of $NH_3$ conversion to ammonia salts in the atmosphere (Chen et al., 2019), which result in excess $NH_3$ remaining in the gaseous phase. Different from China, the reactive N concentrations in some other Asia and Africa countries especially India ($NO_3^-$, $HNO_3$, $NH_3$, and $NH_4^+$: 54%, 46%, 11%, and 94%), South Korea (76%, 42%, 40%, and 9%), Indonesia (21%, 5%, 14%, and 41%), Democratic Congo (9%, 16%, 145%, 41%), and West Africa Coast (4%, 11%, 37%, and 106%) still exhibited stable increases during 2000-2019. The results indicated that no strong reactive N emission control measures were implemented in these countries, which should be further exerted imperatively.

Compared with China, the long-term trends of reactive nitrogen components Europe and the United States were relatively stable. In the Europe, the concentrations of $NO_3^-$ ($NO_3$-N), $HNO_3$, and $NH_3$ exhibited increases during 2000-2007 (0.7%/yr, 2.3%/yr, and 2.1%/yr), while they experienced slight decreases after 2007. The $NH_4^+$ ($NH_4$-N) level displayed continuous decrease since 2000. The result was closely linked with the trends of $NO_x$ and $NH_3$ emissions derived from satellite retrieval (Cooper et al., 2017; Luo et al., 2022). In the United States, both of $NO_3^-$ and $NH_4^+$ showed persistent decreases during 2000-2019. Zhang et al. (2018) have confirmed that $NO_x$ emission in the eastern US has experienced persistent decrease since 1990, which facilitated the decreases of $NO_3^-$ and $NH_4^+$ levels. However, the ambient $HNO_3$ and $NH_3$ concentrations displayed slight increases during 2000-2007 (2.1%/yr), and then remained relatively stable since 2007. Liu et al. (2019) also found similar characteristic of ambient $NH_3$ trend in the United States. In fact, the $NH_3$ concentrations in

the Middle Plain and eastern US still showed increases due to the lack of $NH_3$ emission control
policies as well as the decline in acidic gases (Warner et al., 2017). The reactive N concentrations
in some countries in South America such as Brazil (9%, 0%, 13%, and 34%) and Argentina (10%,
12%, 18%, and 7%) also remained relatively stable because local anthropogenic emission of reactive
N did not show dramatic increases in the past two decades (McDuffie et al., 2020).
3.5 Projection of future ambient reactive nitrogen components
For the future reactive nitrogen component estimates, the ensemble model was applied to
predict the reactive nitrogen component concentrations under the SSP1-2.6, SSP2-4.5, SSP3-7.0,
and SSP5-8.5 scenarios. SSP1-2.6 represents the low emission pathways. In SSP1-2.6, the projected
average $NO_3^-$ concentrations in most countries experienced rapid decreases from 2020 to 2100
(Figure 7 and Table 1). The mean concentrations of $NO_3^-$ in China, India, Europe, and the United
States decreased from $1.16 \pm 0.35$, $1.23 \pm 0.42$, $0.41 \pm 0.14$, and $0.27 \pm 0.09$ μg N m$^{-3}$ to $0.33 \pm 0.10$,
$0.65 \pm 0.21$, $0.10 \pm 0.03$, and $0.06 \pm 0.02$ μg N m$^{-3}$ during 2020-2100 in SSP1-2.6 scenario,
respectively. Besides, the $NO_3^-$ concentrations in many other countries of Africa and South America
such as Brazil (-127%) and Democratic Congo (-162%) also suffered from drastic decreases in this
scenario. SSP2-4.5 scenario represents the middle range of plausible future pathways (Nazarenko
et al., 2022). In this scenario, the predicted average $NO_3^-$ concentrations in China, India, Europe,
and the United States decreased from $1.19 \pm 0.40$, $1.43 \pm 0.35$, $0.44 \pm 0.13$, and $0.24 \pm 0.08$ μg N
m$^{-3}$ to $0.41 \pm 0.14$, $0.95 \pm 0.32$, $0.24 \pm 0.08$, and $0.05 \pm 0.02$ μg N m$^{-3}$ during 2020-2100, respectively.
SSP3-7.0 and SSP5-8.5 denote the less investment in the environment and heavily relies on
traditional energy for rapid economic development, respectively. The ambient $NO_3^-$ in these
scenarios generally showed the higher concentrations compared with other scenarios. For instance,
the $NO_3^-$ concentrations in China reduced from $1.25 \pm 0.40$ (SSP3-7.0) and $1.21 \pm 0.39$ (SSP5-8.5)
to $0.75 \pm 0.25$ (SSP3-7.0) and $0.58 \pm 0.18$ (SSP5-8.5) μg N m$^{-3}$ during 2020-2100 (Figure 8),
respectively. The higher $NO_3^-$ concentrations in SSP3-7.0 and SSP5-8.5 might be associated with
the higher anthropogenic $NO_x$ emission. Compard with SSP1-2.6 and SSP2-4.5, the $NO_3^-$
concentrations in some countries during SSP3-7.0 and SSP5-8.5 scenarios displayed slight increases
from 2020 to 2040. For instance, the ambient $NO_3^-$ concentrations in Indonesia increased by 12%
(SSP3-7.0) and 5% (SSP5-8.5), respectively.
The temporal variations of ambient $HNO_3$ were similar to those of $NO_3^-$ concentrations. The
mean concentrations of $HNO_3$ in China, India, Europe, and the United States decreased from 0.25
$\pm$ 0.09, 0.50 $\pm$ 0.16, 0.18 $\pm$ 0.06, and 0.08 $\pm$ 0.03 $\mu$g N m$^{-3}$ to 0.05 $\pm$ 0.01, 0.24 $\pm$ 0.08, 0.05 $\pm$ 0.02,
and 0.03 $\pm$ 0.01 $\mu$g N m$^{-3}$ during 2020-2100 in SSP1-2.6 scenario (Figure S13-S14 and Table S3),
respectively. However, the decreasing ratios of ambient $HNO_3$ levels especially in some developing
countries such as Democratic Congo (-13%) and West Africa Coast (-47%) were much less than
those of ambient $NO_3^-$ levels. For the SSP3-7.0 and SSP5-8.5 scenarios, the $HNO_3$ levels in some
developing countries such as Democratic Congo (18%), West Africa Coast (16%), Indonesia (13%)
even experienced moderate increases. It was assumed that the government gave less investment in
environment improvement and the anthropogenic emission did not show marked decrease under the
condition of SSP3-7.0 scenario (Chen et al., 2023; Chen et al., 2020).
As shown in Figure S15-S18 and Table S4-S5, the higher ambient $NH_3$ and $NH_4^+$
concentrations also focused on India and North China. In SSP1-2.6, the ambient $NH_3$ ($NH_4^+$)
concentrations in China, India, Europe, and the United States decreased from 3.51 $\pm$ 1.12 (2.00 $\pm$
0.62), 6.30 $\pm$ 2.12 (4.26 $\pm$ 1.42), 1.54 $\pm$ 0.51 (0.75 $\pm$ 0.24), and 1.79 $\pm$ 0.59 (0.53 $\pm$ 0.17) $\mu$g N m$^{-3}$
to 1.75 $\pm$ 0.58 (0.58 $\pm$ 0.19), 2.57 $\pm$ 0.85 (1.25 $\pm$ 0.41), 1.15 $\pm$ 0.36 (0.50 $\pm$ 0.16), and 1.58 $\pm$ 0.52
(0.45 $\pm$ 0.15) $\mu$g N m$^{-3}$ during 2020-2100. Compared with SSP1-2.6, the ambient $NH_3$ and $NH_4^+$
concentrations in heavy-pollution scenarios (SSP3-7.0, and SSP5-8.5) scenarios did not show
marked decreases from 2020-2100. Some developing countries such as Argentina (9%), Democratic
Congo (25%), and West Africa Coast (24%) even suffered from persistent increases of ambient $NH_3$
and $NH_4^+$ levels. It might be associated with ineffective control of $NH_3$ emission compared with
$NO_x$ emission.
3.6 Conclusions and limitations
The ground-level ambient reactive nitrogen observations, satellite retrievals, GEOS-Chem
model output, and many other geographical covariates were integrated into the multi-stage model
to reveal the global patterns of ambient reactive nitrogen components during 2000-2019. Then, these
high-resolution reactive nitrogen dataset during the historical period was then utilized as the
constraint to calibrate the CMIP6 dataset in four scenarios during 2020-2100. The results indicated
the cross-validation (CV) $R^2$ values of four reactive nitrogen species showed satisfied performance

($R^2 > 0.55$). At the spatial scale, four reactive nitrogen components exhibited the higher concentrations in China and India. For the temporal variations, the concentrations of estimated ambient reactive nitrogen components in China experienced persistent increases during 2000-2013, while they suffered from drastic decreases since 2013 except $NH_3$, which might be linked with the impact of clean air policy. However, the concentrations of these species in Europe and the United States remained relatively stable since 2000. In the future scenarios, SSP3-7.0 (traditional energy scenario) and SSP1-2.6 (carbon neutrality scenario) displayed the highest and lowest reactive nitrogen component concentrations, respectively.

Global trends of four reactive nitrogen components during 2000-2100 emphasizes the urgent mitigation measures (carbon neutrality pathway) to reduce precursor emissions in order to decrease the concentrations and depositions of reactive nitrogen components especially in China and India. Furthermore, our result could give valuable insights into the impact of reactive nitrogen components on human health and ecological environment. However, this study still shows some limitations. First of all, the observation networks mainly focus on China, Europe, and the United States, and thus the simulations in many other regions might show large uncertainties. Secondly, both of the GEOS-Chem output and CMIP6 future climate scenario data also exhibits large uncertainties, which could impact the reliability of this study. Lastly, our predictions were performed on the basis of the premise that the world was steadily developing, and cannot predict the impacts of uncontrollable factors (e.g., COVID-19, Russia-Ukraine War).

**Competing interests**

The contact author has declared that none of the authors has any competing interests.

**Acknowledgements**

This work was supported by the National Natural Science Foundation of China (U23A2030), Shanghai Rehabilitation Medical Association health management research project (2023JGKT32), Academic Mentorship for Scientific Research Cadre Project (AMSCP-24-05-03).

**Data availability**

The CMIP6 dataset used in this publication is available in https://esgf.nci.org.au/projects/cmip6-nci/.

**Author contributions**

431     LR, SWW, and WGH designed the study; LR developed the model; GYN, ZLJ, XTZ, and SYB

432     analyzed the observation and model data. LR wrote this manuscript.

433

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

**Figure 1** The workflow of global full-coverage reactive nitrogen estimates during 2000-2100.
GC$_{output}$ denotes the GEOS-Chem output.

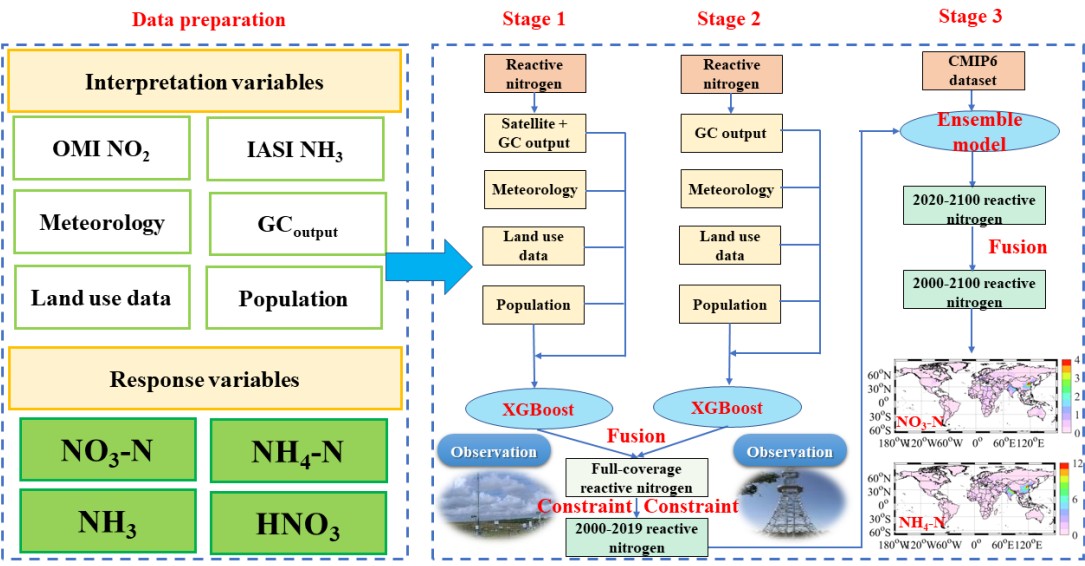


**Figure 2** The predictive performances of four reactive nitrogen components including $NO_3^-$ ($NO_3$-
N) (a), $HNO_3$ (b), $NH_3$ (c), and $NH_4^+$ ($NH_4$-N) (d). The model was constructed with 90% original
data and the remained data was applied to validate the model. The black solid line denotes the best-
fitting curve for all of the points, while the black dashed line represents the diagonal, which means
the same observed and simulated values. The color scale denotes the sample size.

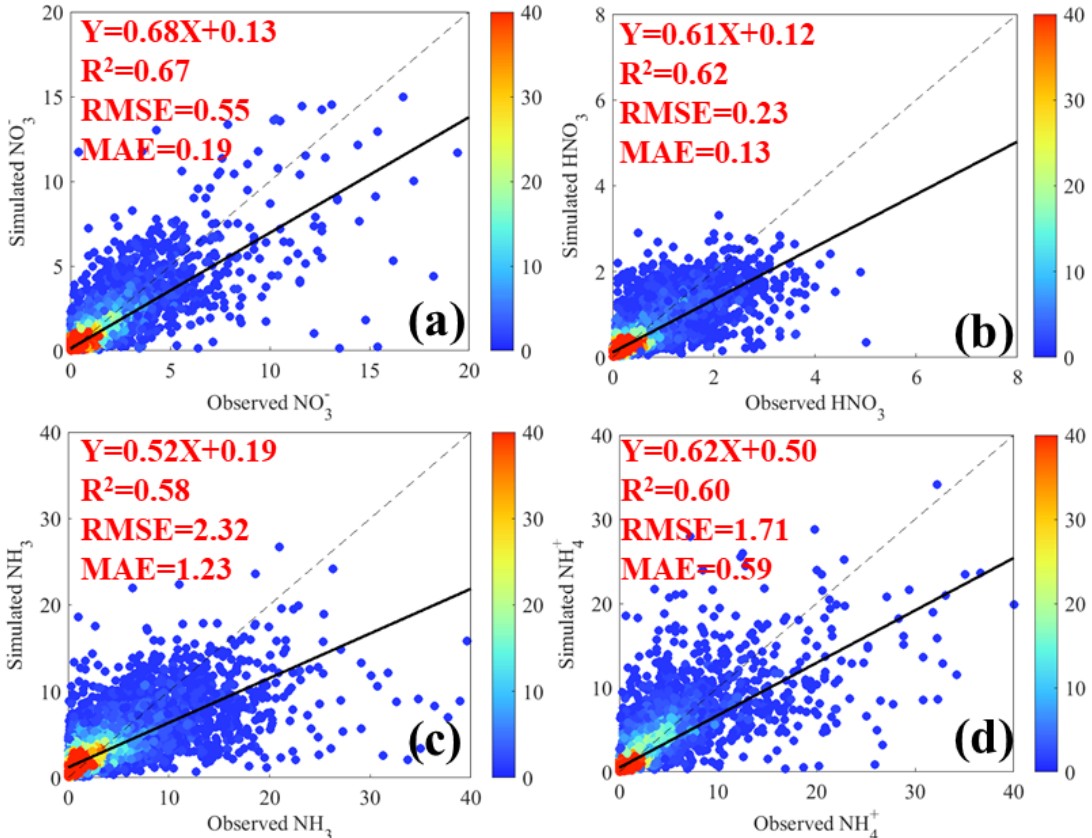


**Figure 3** The spatiotemporal variations of $NO_3^-$ ($NO_3$-N), $HNO_3$, $NH_3$, and $NH_4^+$ ($NH_4$-N)
concentrations in East Asia (a-d) (Unit: μg N m$^{-3}$).

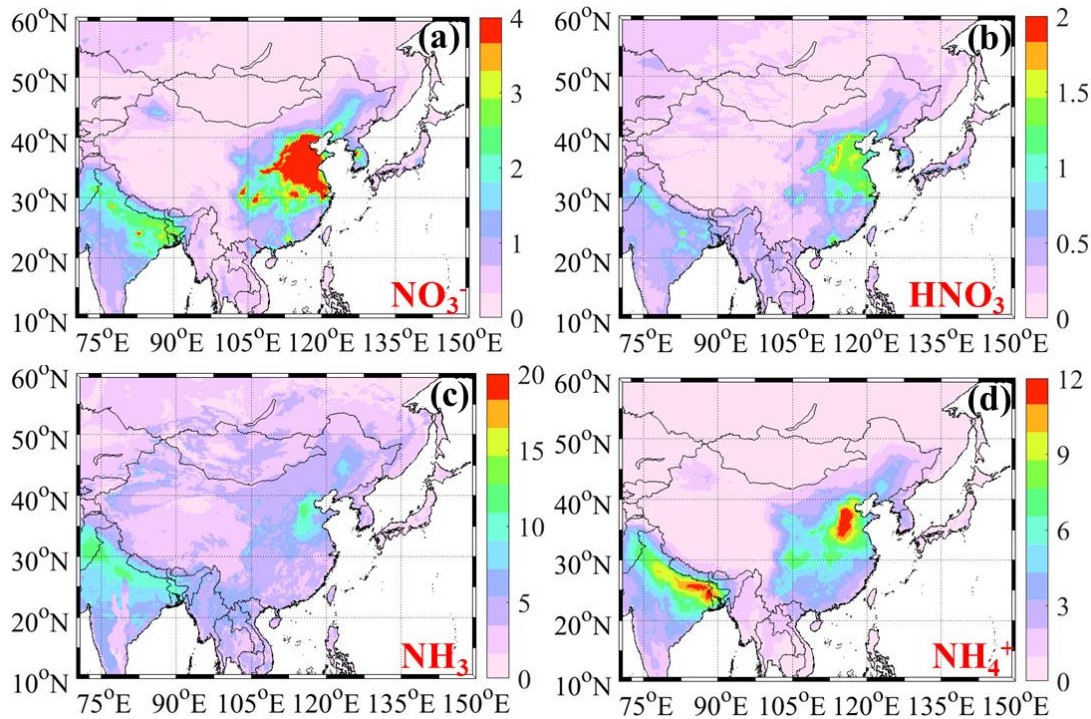


**Figure 4** The spatiotemporal variations of $NO_3^-$ ($NO_3$-N), $HNO_3$, $NH_3$, and $NH_4^+$ ($NH_4$-N)
concentrations in Europe (a-d) (Unit: µg N m$^{-3}$).

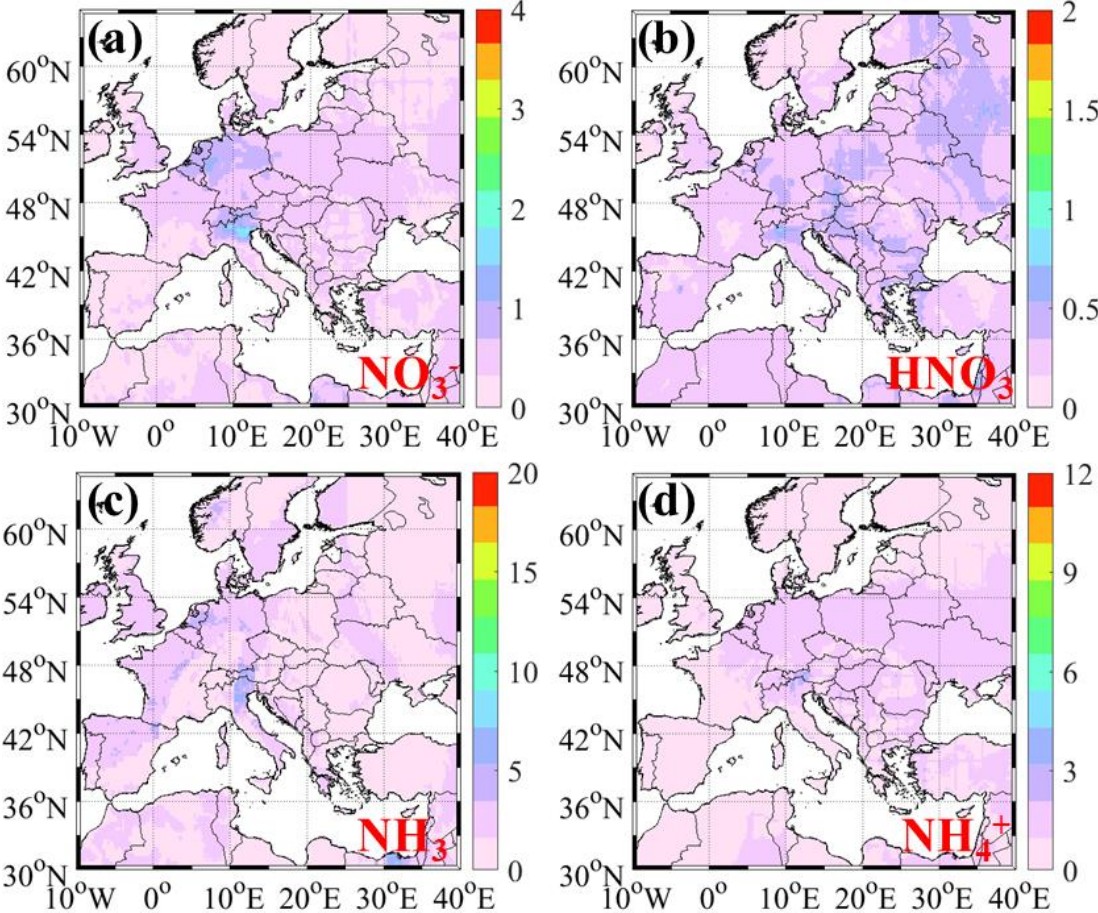


**Figure 5** The spatiotemporal variations of $NO_3^-$ ($NO_3$-N), $HNO_3$, $NH_3$, and $NH_4^+$ ($NH_4$-N)
concentrations in North America (a-d) (Unit: μg N m$^{-3}$).

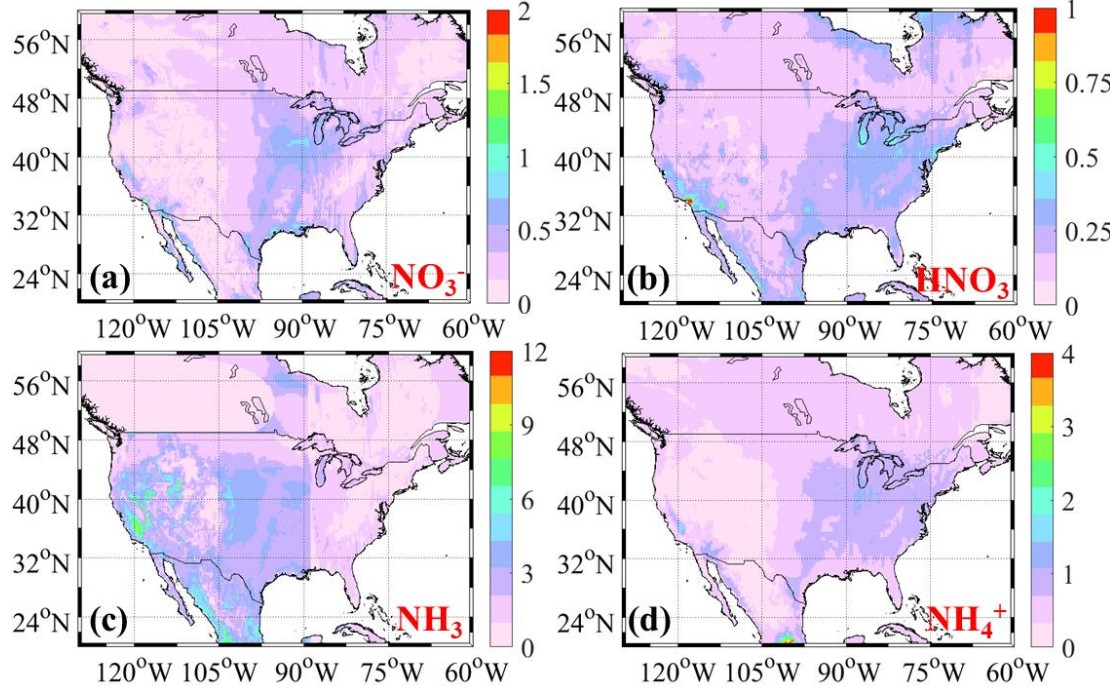


**Figure 6** The long-term variations of $NO_3^-$ ($NO_3$-N), $HNO_3$, $NH_3$, and $NH_4^+$ ($NH_4$-N) concentrations
in China (pink), Europe (green), and the United States (cyan) (Unit: µg N m$^{-3}$).

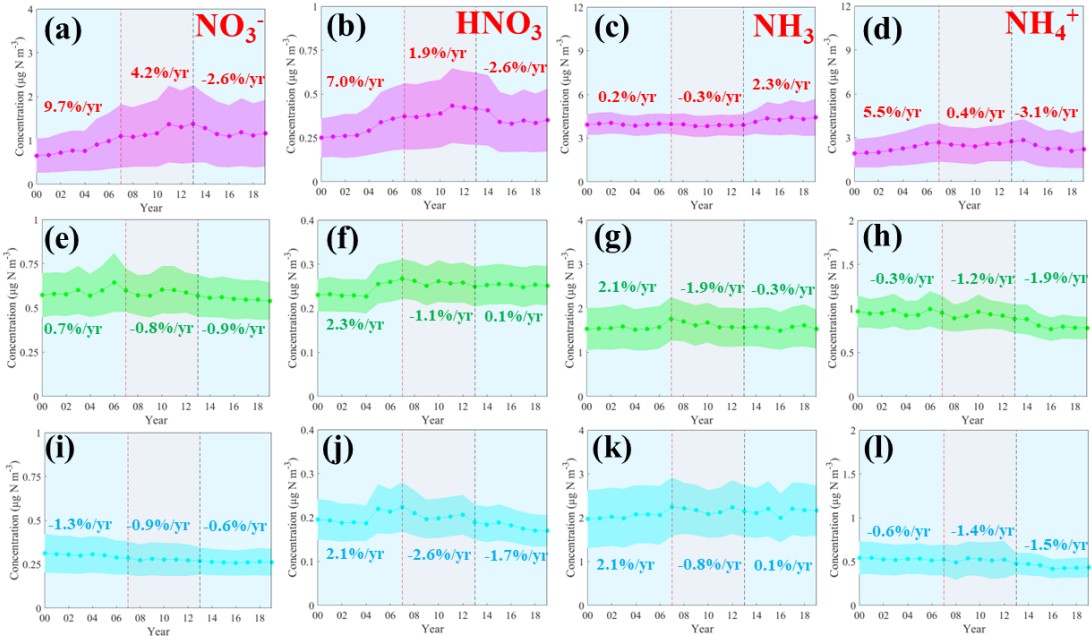


**Figure 7** Spatial variations of projected global ambient concentrations of reactive nitrogen components under different climate change scenarios (Unit: μg N m$^{-3}$). Panels (a-b) represent the annual mean concentrations of ambient NO$_3^-$ (NO$_3$-N) under SSP1-2.6, SSP2-4.5 during 2021-2100, respectively.

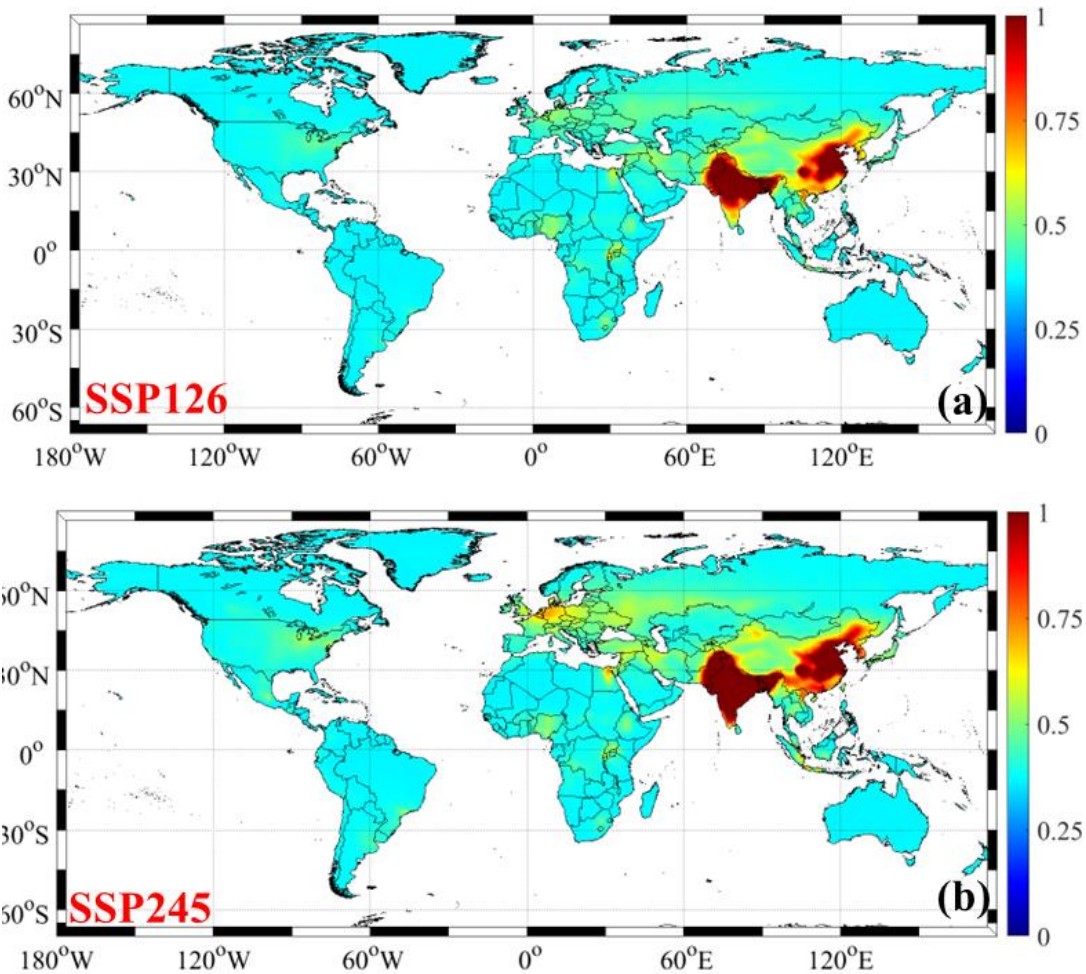

**Figure 8** Spatial variations of projected global ambient concentrations of reactive nitrogen components under different climate change scenarios (Unit: μg N m⁻³). Panels (a-b) represent the annual mean concentrations of ambient $NO_3^-$ ($NO_3$-N) under SSP3-7.0, and SSP5-8.5 during 2021-2100, respectively.

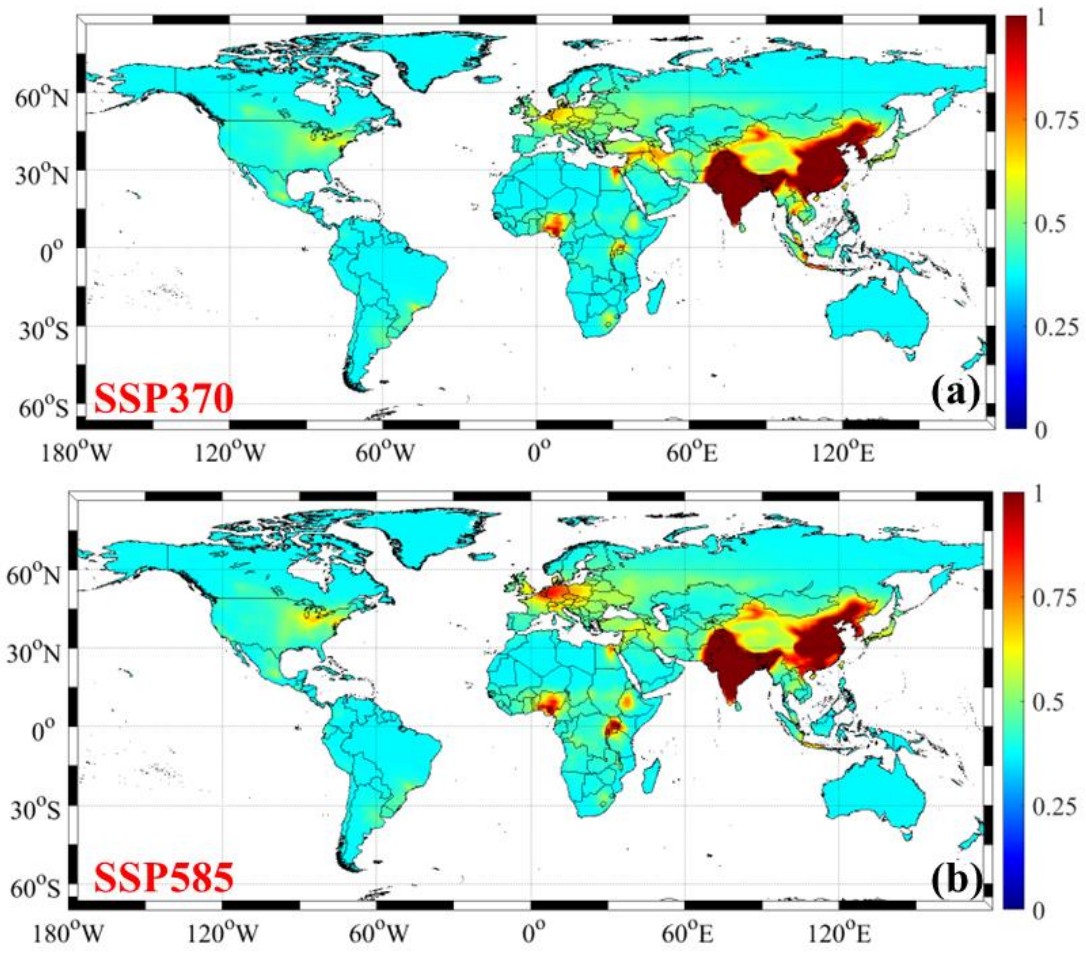

**Table 1** The temporal variations of ambient $NO_3^-$ ($NO_3$-N) concentrations (average concentrations,

 Unit: µg N m$^{-3}$) in selected countries during 2000-2100.

| Scenario | $NO_3^-$ | China | India | Europe | United States | Brazil | Argentina | Democratic Congo | West Africa Coast | Indonesia | South Korea |
|---|---|---|---|---|---|---|---|---|---|---|---|
| Historical | 2000 | 0.66 | 0.95 | 0.57 | 0.31 | 0.29 | 0.21 | 0.56 | 0.30 | 0.31 | 1.06 |
|  | 2005 | 0.91 | 1.26 | 0.60 | 0.30 | 0.31 | 0.22 | 0.56 | 0.30 | 0.34 | 1.38 |
|  | 2010 | 1.17 | 1.53 | 0.60 | 0.28 | 0.33 | 0.23 | 0.58 | 0.31 | 0.32 | 1.43 |
|  | 2013 | 1.39 | 1.63 | 0.57 | 0.27 | 0.30 | 0.24 | 0.58 | 0.31 | 0.33 | 1.57 |
|  | 2015 | 1.16 | 1.34 | 0.56 | 0.26 | 0.32 | 0.22 | 0.58 | 0.32 | 0.45 | 1.88 |
|  | 2019 | 1.18 | 1.46 | 0.54 | 0.26 | 0.32 | 0.23 | 0.61 | 0.32 | 0.37 | 1.87 |
| SSP1-2.6 | 2020 | 1.16 | 1.23 | 0.41 | 0.27 | 0.25 | 0.18 | 0.55 | 0.28 | 0.34 | 1.81 |
|  | 2040 | 0.82 | 1.12 | 0.28 | 0.10 | 0.21 | 0.14 | 0.62 | 0.32 | 0.28 | 1.46 |
|  | 2060 | 0.69 | 1.01 | 0.15 | 0.05 | 0.18 | 0.13 | 0.43 | 0.25 | 0.24 | 0.69 |
|  | 2080 | 0.41 | 0.89 | 0.12 | 0.05 | 0.17 | 0.12 | 0.32 | 0.17 | 0.20 | 0.39 |
|  | 2100 | 0.33 | 0.65 | 0.10 | 0.06 | 0.11 | 0.08 | 0.21 | 0.11 | 0.16 | 0.23 |
| SSP2-4.5 | 2020 | 1.19 | 1.43 | 0.44 | 0.24 | 0.26 | 0.19 | 0.52 | 0.29 | 0.37 | 1.85 |
|  | 2040 | 1.09 | 1.35 | 0.43 | 0.16 | 0.22 | 0.16 | 0.57 | 0.31 | 0.35 | 1.80 |
|  | 2060 | 0.89 | 1.22 | 0.35 | 0.11 | 0.20 | 0.15 | 0.51 | 0.27 | 0.32 | 1.25 |
|  | 2080 | 0.63 | 1.06 | 0.29 | 0.07 | 0.17 | 0.12 | 0.42 | 0.23 | 0.23 | 0.68 |
|  | 2100 | 0.41 | 0.95 | 0.24 | 0.05 | 0.14 | 0.10 | 0.36 | 0.19 | 0.20 | 0.38 |
| SSP3-7.0 | 2020 | 1.25 | 1.59 | 0.53 | 0.33 | 0.31 | 0.22 | 0.64 | 0.34 | 0.42 | 1.95 |
|  | 2040 | 1.36 | 1.50 | 0.47 | 0.24 | 0.26 | 0.19 | 0.61 | 0.33 | 0.47 | 1.89 |
|  | 2060 | 1.18 | 1.35 | 0.42 | 0.19 | 0.22 | 0.16 | 0.56 | 0.30 | 0.41 | 1.56 |
|  | 2080 | 0.96 | 1.15 | 0.36 | 0.16 | 0.18 | 0.13 | 0.54 | 0.29 | 0.35 | 1.35 |
|  | 2100 | 0.75 | 1.08 | 0.33 | 0.12 | 0.15 | 0.11 | 0.51 | 0.27 | 0.30 | 1.24 |
| SSP5-8.5 | 2020 | 1.21 | 1.50 | 0.53 | 0.28 | 0.28 | 0.20 | 0.57 | 0.31 | 0.37 | 1.91 |
|  | 2040 | 1.28 | 1.42 | 0.49 | 0.23 | 0.25 | 0.18 | 0.60 | 0.32 | 0.39 | 1.85 |
|  | 2060 | 1.05 | 1.30 | 0.47 | 0.24 | 0.20 | 0.15 | 0.55 | 0.29 | 0.35 | 1.44 |
|  | 2080 | 0.86 | 1.10 | 0.44 | 0.25 | 0.15 | 0.11 | 0.50 | 0.27 | 0.29 | 1.26 |
|  | 2100 | 0.58 | 1.02 | 0.31 | 0.20 | 0.13 | 0.09 | 0.46 | 0.25 | 0.25 | 1.05 |

673
674