# Peer review of "2000-2100 based on the multi-stage model"

_EGUsphere, 2024_

## Author Comment (AC2)

Dear editor,

Thank for editor's and reviewer's suggestions. We have corrected some errors in the revised version based on reviewer's suggestions from point to point. The detailed revisions have been shown in the revised manuscript.

**#RC 1**

Li et al. developed a multi-stage model to predict the reactive nitrogen concentrations during 2000-2100. Overall, the topic is very interesting and the manuscript fits the scope of ACP. However, the manuscript still suffers from some major flaws especially some sections lacks of necessary examination and discussion. I recommend the manuscript for publication on ACP after the following comments have been well addressed.

**Response:** Thank for reviewer's suggestions. We have significantly revised the manuscript based on reviewer's suggestions. The detailed revisions are as follows:

**Comment 1:** "Ambient" should be added between "of" and "reactive" because the authors simulated the ambient reactive nitrogen concentrations.

**Response:** I agree with reviewer's suggestions. The "ambient" has been added between "of" and "reactive".

**Comment 2:** Line 60-69: The introduction about reactive nitrogen estimates in future scenarios is too simple. I suggest the authors must point out the importance and novelty of the simulation in the future scenario.

**Response:** I agree with reviewer's suggestions. The future prediction of reactive nitrogen is also important because these components in the future scenarios could significantly affect the land carbon cycle and greenhouse gas emissions, both of which could aggravate the global climate change and affect the earth system safety.

**Comment 3:** Line 94-102: More detailed GEOS-Chem scheme should be introduced.

**Response:** Thank for reviewer's suggestions. (Line 93-107) I have added more detailed GEOS-Chem scheme in the revised version.

**Comment 4:** Why do you choose XGBoost and LightGBM among all of the decision tree models? Please add the introduction of hyperparameters of these models.

**Response:** Thank for reviewer's suggestions. XGBoost and LightGBM models showed the better predictive accuracy in air pollutant estimates. Moreover, both of these models showed the higher calculation efficiency compared with random forest. Therefore, we choose these models in our study. The hyperparameters of these models have been added in the revised version (Line 164-170).

**Comment 5:** Section 3.1: How about the spatiotemporal transferability of this multi-stage model? The authors must add some tests about it.

**Response:** Thank for reviewer's suggestions. We have added the transferability test in the revised manuscript. We used the 90% data as the training samples to train the model, and the remained one was regarded as the test samples. For instance, we used the ground-level observations in 2000-2017 to train the model, and the data during 2018-2019 was utilized to examine the model. After 10 rounds, we could obtain the final predictive performance. The results suggested the transferability CV $R^2$ values for four species was close to the original CV $R^2$ values. Therefore, the model could be treated to be robust.

[Figure]

**Comment 6:** Line 223-225: I cannot agree with this view because many croplands were also distributed on Southeast China.

**Response:** Thank for reviewer's suggestions. We have corrected these errors.

**Comment 7:** The historical trends of reactive nitrogen concentrations in some other regions such as Africa and South America could be also introduced simply because the authors only focused on China, Europe, and the United States.

**Response:** Thank for reviewer's suggestions. We have added some discussions about spatial and temporal variations of reactive N concentrations in Africa and South America (Section 3.2-3.4)

**Comment 8:** Section 3.5: This part is just result rather than discussion. The author must add some discussions about the future trends of reactive nitrogen species.

**Response:** Thank for reviewer's suggestions. (Line 350-394) We also have added some discussions about the future trends of reactive N species.

**Comment 9:** Section 3.6: The GEOS-Chem output also shows some uncertainties.

**Response:** I agree with reviewer's suggestions. I have added the limitations in the revised version.

**Comment 10:** Section 3.6: The authors should add the conclusions rather than the implications alone.

**Response:** I agree with reviewer's suggestions. I have added the conclusions in the revised version.

**#RC 2**

The manuscript of egusphere-2024-69 entitled "Global estimates of reactive nitrogen components during 2000-2100 based on the multi-stage model" presented the evaluation of reactive nitrogen at the current status and projected to the future using data fusion with CMIP6 results. This kind of topic should be clarified and I can fully understand the importance of this study. However, many parts need to be amended. Especially, as a major comment to this manuscript, I do not fully follow why the main discussion has been focused on only the U.S., Europe, and China. Because the validations were based on NNDMN over China, CASTNET in the U.S., EMEP over Europe, and EANET over Asia, these areas need to be fully discussed, but I do not find the detailed discussion over Asia, and the status over only China seems to be focused. Moreover, other areas could be precisely estimated based on the fusion of satellite data and can be discussed, but why the status of India was sometimes discussed? Toward the dramatic change in the 2100s, the targeted areas

regarding reactive nitrogen can be discussed through this study, and I expect such perspective discussions. In addition, the discussion for each region and each species will be discussed in longer sentences in general, so how about summarizing as one or two table(s)? Taking into consideration the importance of the findings through this study, I would like to request a clear presentation. Please also see and address the following specific comments before the possible publication from the journal of ACP.

**Response:** Thank for reviewer's suggestions. We have added many discussions in many other regions (hotspots of reactive nitrogen) such as Africa, South America, and other countries in Asia (except China) (Section 3.4 and 3.5). The spatial, long-term trends of reactive nitrogen concentrations in these countries were discussed (Section 3.4 and 3.5). Besides, we also discussed the long-term trends from 2000 to 2100 based on region or country and analyze the potential factors. We have shown the long-term trends based on some tables (Table 1 and S3-S5).

**Specific comments:**

**Comment 1:** Title and abstract: I do not fully follow the intention of "stage" in the manuscript title itself. This point can be followed from Section 2.3, so it will be better to clarify this methodology in the abstract.

**Response:** Thank for reviewer's suggestions. We pointed out the stage in the title because we developed the multi-stage model, which could ensure the higher accuracy of ambient reactive nitrogen estimates. I think the development of this model is an important novelty in our study. I agree with reviewer's suggestions. We have further clarified the methodology in the abstract.

**Comment 2:** Page 2, Line 31-32: What are the direct emissions for $HNO_3$? The oxidation pathway from $NO_x$ will be the source of $HNO_3$. Please clarify.

**Response:** Thank for reviewer's suggestions. (Line 31-34) I have corrected these errors. $HNO_3$ was mainly generated from the oxidation of $NO_x$.

**Comment 3:** Page 3, Line 60-62: I am confused reactive nitrogen themselves affect the land carbon cycle and greenhouse gas emissions, or vice versa.

**Response:** Thank for reviewer's suggestions. The deposition of reactive nitrogen could affect the soil $CO_2$, $CH_4$, and $N_2O$ emissions and land carbon cycle (increase or decrease carbon sink), which has been demonstrated by many previous studies (Reay et al., NG, 2008; Lu et al., PNAS, 2021).

**Comment 4:** Page 3, Line 70-73: Why only four components of $NO_3^-$, $HNO_3$, $NH_3$, and $NH_4^+$ is regarded as important species? How about $NO_x$ (NO and/or $NO_2$) and PAN? The appropriate references are required to support this target.

**Response:** Thank for reviewer's suggestions. Indeed, ambient reactive nitrogen possesses many species. In our study, we only choose four important reactive components to estimate. We choose these species because they showed sufficient ground-level observations and satellite products, which could ensure the robustness of estimate. As the companion paper, we also submitted a manuscript about global $NO_2$ estimates to other journals because a single paper cannot introduce the spatiotemporal variations of too many species. Furthermore, $NO_2$ generally shows more ground-level observations, we developed other model to predict the full-coverage $NO_2$ concentrations at the global scale. Therefore, we did not incorporate the $NO_2$ estimates in this study. For PAN, only satellite products instead of ground-level observations cannot fully ensure the predictive accuracy of global PAN estimates. Furthermore, the lack of field measurement makes us difficult to use the machine-learning model. Thus, in the future work, we need to use other methods to capture the global PAN concentrations.

**Comment 5:** Page 5, Line 139 and Line 143: Were the input datasets unified or different at the first and second stages? The wording "meteorological factors" and "meteorological parameters", "land use type" and "land use data" impressed us with the different datasets. So, please clarify the difference. If the same dataset were used at these two stages, the expression should be unified.

**Response:** Thank for reviewer's suggestions. These words showed the same implication. We have kept the coordinate to these words in the revised version.

**Comment 6:** Page 6, Line 147-149: At the final stage, did the authors use the CMIP6 dataset from 16 earth system model as an ensemble average? Please clarify.

**Response:** Thank for reviewer's suggestions. Indeed, we use the CMIP6 dataset from 16 earth system model as an ensemble average.

**Comment 7:** Page 7, Line 178: I do not fully follow how the GEOS-Chem 2.5 degrees dataset was resampled into 0.5 degrees. Please add more explanations for "appropriate algorithms" explained here.

**Response:** Thank for reviewer's suggestions. (Line 186-187) We used the Kriging interpolation to resample the GEOS-Chem output from $2.5°$ to $0.25°$. Although the Kriging interpolation might show some uncertainties, the grid-based Kriging interpolation shows less uncertainties compared with site-based Kriging because the grid is evenly distributed at the global scale.

**Comment 8:** Page 7, Line 180-183: I can understand this approach, but what kind of variables were removed during the model development process? Such additional information would be helpful for readers, so please consider including them in the supplemental information.

**Response:** Thank for reviewer's suggestions. In our study, some redundant variables including the areas of snow/ice and tundra were removed in the final model, which has been explained in the supporting information.

**Comment 9:** Page 7, Line 190 and 191: Even though such statistical metrics have been well used in the model evaluation papers, the abbreviations for RMSE and MAE should be introduced, and defined within the manuscript (or, refer to appropriate references).

**Response:** Thank for reviewer's suggestions. We have added the full names and equations of RMSE and MAE in the supporting information (Page 2 in the SI). In addition, the appropriate references were also added in the revised version.

**Comment 10:** Figure 1: "$NO_3$-N" and "$NH_4$-N" should be unified into the main manuscript. From section 2.3, I feel LightGBM method was applied at the first stage whereas this Fig. 1 seems to be using XGBoost. Please confirm this figure.

**Response:** Thank for reviewer's suggestions. It is our fault, we used XGBoost model in the first stage, and we have corrected the error in the manuscript. Besides, we also unified the "$NO_3$-N" and "$NH_4$-N" throughout the manuscript.

**Comment 11:** Figures 3, 5, 6, 7, and 8: The size of these figures is too small to see the details of structure over the world. Please revise.

**Response:** Thank for reviewer's suggestions. We have reorganized these figures.

Technical corrections:

**Comment 12:** Page 4, Line 88: The monitoring site numbers are needed for EANET as is the same manner for EMEP and CASTNET.

**Response:** Thank for reviewer's suggestions. We have introduced the site number for EANET. The dataset of reactive nitrogen components in other countries of East Asia during 2000-2019 were download from the Acid Deposition Monitoring Network in East Asia (EANET), which includes 41

sites.

---

## Author Response (AR2)

Dear editor,

Thank for editor's and reviewer's suggestions. We have corrected some errors in the revised version based on reviewer's suggestions from point to point. The detailed revisions have been shown in the revised manuscript.

**#RC 1**

The revised manuscript of egusphere-2024-69 has been well addressed in my comments. I would like to agree with the publication from the journal of ACP. Before the final production, please consider including the following two minor points.

**Response:** Thank for reviewer's suggestions. I have significantly revised the manuscript based on reviewer's suggestions.

**Comment 1:** From Fig. 2 to Fig. 8, the caption and legend should be also expressed as "$NO_3$-N" and "$NH_4$-N" to be consistent with the main text.

**Response:** Thank for reviewer's suggestions. We have revised the caption and legend from Fig. 2 to Fig. 8.

**Comment 2:** In Table 1, "many countries" seems to be not a scientifical expression, so how about "selected countries"?

**Response:** Thank for reviewer's suggestions. We have replaced the "many countries" by "selected countries" in Table 1.